# Electrochemical Oxidation of Landfill Leachate after Biological Treatment by Electro-Fenton System with Corroding Electrode of Iron

**DOI:** 10.3390/ijerph19137745

**Published:** 2022-06-24

**Authors:** Juan Tang, Shuo Yao, Fei Xiao, Jianxin Xia, Xuan Xing

**Affiliations:** 1Department of Environmental Science, College of Life and Environmental Science, Minzu University of China, Beijing 100081, China; 21302371@muc.edu.cn (J.T.); jxxia@vip.sina.com (J.X.); 2China Energy Conservation and Environmental Protection Group, Beijing 100082, China; yaoshuo@cecep.cn (S.Y.); xiaofei@cecep.cn (F.X.)

**Keywords:** landfill leachate, electro-Fenton oxidation, response surface methodology, boron-doped diamond, corroding electrode of iron

## Abstract

Electrochemical oxidation of landfill leachate after biological treatment by a novel electrochemical system, which was constructed by introducing a corroding electrode of iron (Fe_c_) between a boron-doped diamond (BDD) anode and carbon felt (CF) cathode (named as BDD–Fe_c_–CF), was investigated in the present study. Response surface methodology (RSM) with Box–Behnken (BBD) statistical experiment design was applied to optimize the experimental conditions. Effects of variables including current density, electrolytic time and pH on chemical oxygen demand (COD) and ammonia nitrogen (NH_3_-N) removal efficiency were analyzed. Results showed that electrolytic time was more important than current density and pH for both COD and NH_3_-N degradation. Based on analysis of variance (ANOVA) under the optimum conditions (current density of 25 mA·cm^−2^, electrolytic time of 9 h and pH of 11), the removal efficiencies for COD and NH_3_-N were 81.3% and 99.8%, respectively. In the BDD–Fe_c_–CF system, organic pollutants were oxidized by electrochemical and Fenton oxidation under acidic conditions. Under alkaline conditions, coagulation by Fe(OH)_3_ and oxidation by Fe(VI) have great contribution on organic compounds degradation. What is more, species of organic compounds before and after electrochemical treatment were analyzed by GC–MS, with 56 kinds components detected before treatment and only 16 kinds left after treatment. These results demonstrated that electrochemical oxidation by the BDD–Fe_c_–CF system has great potential for the advanced treatment of landfill leachate.

## 1. Introduction

Municipal solid waste is growing rapidly worldwide. Based on the reports of the World Bank, the amount of municipal solid waste is about 2 billion tons each year and will increase to 3.4 billion tons by 2050 [1]. The China Statistical Yearbook showed that 22.8 million tons of waste was produced in China in 2018. Sanitary landfill, as the terminal processing mode of household garbage, has been widely used in solid waste disposal [2]. In the United States, about 60~70% solid waste was disposed in sanitary landfills. In China, about 11.8 million tons of solid waste was treated in sanitary landfills each year. However, huge volumes of landfill leachate with recalcitrant contaminants are produced. According to previous study, about 70–100 L landfill leachate for per ton of solid waste are formed through internal water in garbage, biochemical reactions and atmospheric precipitation [3]. The leachate contains large amounts of dissolved organic compounds, hazardous chemicals, soluble salts, heavy metals with high chemical oxygen demand (COD), ammonia nitrogen (NH_3_-N) concentrations and low biochemical oxygen demand (BOD)/COD ratios. What is more, the properties of landfill leachate may change according to the parameters of moisture content, temperature and waste composition, etc. [4]. Landfill leachate is toxic and can induce contamination into surface water, groundwater and soil, which has great potential to threaten surrounding environment and ecosystems [5]. Therefore, effective and economical methods for its treatment are urgent.

The complex composition of landfill leachate makes it difficult to be purified. In recent years, biological treatment process combined with membrane separation have been widely applied for landfill leachate commercialized treatment. The biodegradable compounds are removed in biological process and then the non-biodegradable compounds are removed by following membrane separation processes to make sure the effluence meets the discharge standard. However, the cost of membrane separation is high, and a lot of concentrated landfill leachate is produced which needs further treatment. Hence, numerous studies have investigated this problem in order to lower the cost and prevent leachate concentrate production during leachate treatment. Advanced oxidation processes (AOPs) using hydroxyl radicals (·OH) as oxidants have been identified as an economically effective way to mineralize the organic components [1,6,7,8,9]. Among them, the electrochemical oxidation method has attracted more attention for bio-refractory organic pollutants degradation because of its high efficiency, mild operation conditions and environmental compatibility. Electrode materials are the key factor for electrochemical oxidation process [10]. Among all kinds of anode materials, the boron-doped diamond (BDD) anode with its high stability, wide potential window and extremely high oxygen evolution properties has been regarded as one of the best electrodes [11,12,13,14,15,16]. What is more, the electro-Fenton (EF) system constructed with BDD anode and carbon fiber (CF) cathode has attracted much more attention because of its high efficiency and economic benefits [16,17,18]. The basic principles of the EF system are shown in Equations (1) and (2):O_2_ + 2H^+^ + 2e^−^ → H_2_O_2_(1)
Fe^2+^ + H_2_O_2_ + H^+^ → Fe^3+^ + H_2_O + ·OH(2)

The oxidation ability of the BDD-CF system is significantly enhanced and has been applied in organic compounds degradation, such as surfactants, herbicides, dyes and endocrine disrupting chemicals [19,20,21]. However, there are still some drawbacks hampering its application, such as the narrow pH value, which is usually controlled around 3, high cost of ferrous salts and large amounts of iron mud [22]. According to a previous report by our group [22], inserting a sheet of iron between a BDD anode and CF cathode can construct a BDD–Fe_c_–CF system. Fe^2+^ can be released form corroding electrode of iron under the affection of electronic fields and solvent environments. Under acidic conditions, Fe^2+^ can catalyze H_2_O_2_ formed at the CF cathode and remove organic compounds efficiently. Under alkaline conditions, Fe(OH)_2_ and Fe(OH)_3_ were formed to remove organic compounds by coagulation. The BDD–Fe_c_–CF system was able to remove organic compounds with high efficiency in a wide pH value of 3~11 and has great potential for practical application.

In the present study, the BDD–Fe_c_–CF system was applied to the treatment of biological effluents of landfill leachate for COD and NH_3_-N removal. The optimum conditions are investigated by response surface methodology (RSM) with Box–Behnken (BBD) statistical experiment design that does not require the installation of an axial point and abundant continuous tests [23,24,25]. The influence of variables, including current density (*X_1_*), electrolytic time (*X_2_*) and pH (*X_3_*), on COD removal efficiency ($Y_{COD}$) and NH_3_-N removal efficiency ($Y_{NH3-N}$) have been analyzed. Furthermore, the organic components before and after electrochemical treatment were identified and compared by GC–MS.

## 2. Materials and Methods

### 2.1. Landfill Leachate Samples and Chemicals

Landfill leachate samples were taken from the effluent after biological treatment process in ASUWEI municipal solid waste landfills (Beijing). Samples were kept in polyethylene bottles in a refrigerator at 4 °C [12]. Features of landfill leachate samples after biological treatment process are shown in Table 1. All the chemicals were of analytical grade and purchased from the Chemical Reagent Co., Ltd., of the traditional Chinese Medicine Group. Ultrapure water was supplied by Milli-Q water (≥18.2 MΩ cm^−1^). BDD anode (Condias GmnH, Itzehoe, Germany) with a size of 20 mm × 20 mm × 1 mm was used as the anode. CF (Sigma-Aldrich, St. Louis, MS, USA) of the same size was used as the cathode. The corroding electrode of iron (Fe_c_) was bought from Liduboyi Technologies, Beijing, China. Before electrolysis, the CF cathode was soaked in NaOH (4.5 M) and HCl (5 M), separately, for 10 min and washed by ultrapure water to neutral. Fe_c_ was first sanded by fine sandpaper, then soaked in HCl (0.5 M) for 10 min and finally washed by ultrapure water three times before use.

### 2.2. Electrochemical System

The bulk oxidation of landfill leachate was performed in a one-compartment cell under galvanostatic conditions (20 mA cm^−2^) at room temperature (20 °C) in a 400 mL beaker. The 250 mL solution was stirred by a magnetic stirring bar during the electrolysis process. Constant direct current was provided by a DC voltage stabilizing power supply (Beijing Dahua DH1765-1). The BDD anode and CF cathode are both 4 cm^2^. The Fe_c_ of the same size was inserted between the anode and cathode without an electronic charge. The gaps between the anode and cathode were set to 15 mm. Samples were collected from the cell at various intervals for chemical analysis. The glass reactor was placed in an ultrasonic cleaning instrument for at least 30 min and cleaned by ultrapure water three times to remove the organic compounds sufficiently. 

### 2.3. Analytical Methods

COD of the solutions were measured by the titrimetric method using dichromate as the oxidant in acidic solution at 150 °C for 2 h with a COD digestion instrument (INESA, Shanghai, China). COD removal efficiency was calculated by:(3)$$D_{COD}=\frac{COD_{0}-COD_{t}}{COD_{0}}\times 100\%$$
where *COD*_0_ is the initial concentration and *COD_t_* is the concentration after electrolysis time *t* (h).

NH_3_-N was measured by Nessler’s reagent colorimetry with a portable water quality analyzer (HANNA HI-96733, Italy). H_2_SO_4_ (5 M) and NaOH (5 M) were used to adjust pH of the landfill leachate by INESA PHS-3C. NH_3_-N removal efficiency was calculated by:(4)$$D_{\mathrm{NH}3-N}=\frac{C_{0}-C_{t}}{C_{0}}\times 100\%$$
where *C_0_* and *C_t_* were concentrations of NH_3_-N (mg L^−1^) at intimal time of *t*_0_ and after electrolysis time of *t* (h), respectively.

The landfill leachate before and after electrochemical oxidation was analyzed by gas chromatography (GC) (Agilent 6890; Agilent Technologies, Palo Alto, CA, USA) and mass spectrometry (MS) (Agilent 5973). DP-5MS capillary column (0.25 mm × 0.25 μm × 30 m) was employed for GC separation. The GC equipment was operated in a temperature programmed mode with an initial temperature of 50 °C held for 2 min, then ramped to 290 °C with a 10 °C min^−1^ rate and held for 11 min. The injector and transfer-line temperatures were 280 °C and 250 °C, respectively. The injector was in split mode with the split ratio as 10, and with a 1 μL injection volume. EI source with a temperature of 210 °C and scan field of 45–800 Da. Samples for GC–MS analysis were prepared by following the procedure reported by Lei et al. [26]. A 250 mL leachate sample was initially extracted with CH_2_Cl_2_ (HPLC grade) under neutral conditions, then in alkaline condition (pH 12) by adding drops of NaOH solution and then in acidic condition (pH 2) by adding H_2_SO_4_ using separating funnel. Each extraction was performed twice with 30 mL of CH_2_Cl_2_. The combined extract was dehydrated by rotary evaporator and concentrated using a termovap sample concentrator. 

### 2.4. Response Surface Methodology 

RSM was utilized to optimize the operating conditions and analyze the effects of different operating parameters on electrochemical treatment of landfill leachate taken from biological treatment process. A three-factor, three-level BBD design was used to determine the combinations of input parameters in the program. The current density (*X*_1_), electrolytic time (*X*_2_) and pH value (*X*_3_) were selected as independent variables. COD removal efficiency (*Y*_1_) and NH_3_-N removal efficiency (*Y*_2_) were chosen as output variables. The independent variables and their experimental ranges are shown in Table 2. In order to obtain statistical calculations, the variables *X_i_* were coded in normalized form as *x_i_* ∈ [–1,1], according to:(5)$$x_{i}=\frac{X_{i}-X_{0}}{\delta X}$$
where *X_0_* is the value of *X_i_* at the center point and *δX* is the step change. The experimental levels for each variable were selected based on preliminary experimental results which have been shown in Table 1.

Experimental data were analyzed by Design Expert software (8.0.6) and fitted to a second-order polynomial model:(6)$$Y=b_{0}+b_{1}x_{1}+b_{2}x_{2}+b_{3}x_{3}+b_{11}x_{1}{}^{2}+b_{22}x_{2}{}^{2}+b_{33}x_{3}{}^{2}\;+b_{12}x_{1}x_{2}+b_{13}x_{1}x_{3}+b_{23}x_{2}x_{3}$$
where *Y* is the response variable; *b*_0_ is constant; *b*_1_, *b*_2_ and *b*_3_ are regression coefficients for linear effects; *b*_11_, *b*_22_ and *b*_33_ are quadratic coefficients and *b*_12_, *b*_13_ and *b*_23_ are interaction coefficients. In this study, COD and NH_3_-N removal efficiencies were selected as response variables [16,22].

## 3. Results and Discussion

### 3.1. RSM Analysis

The scheme of the BDD–Fe_c_–CF electrochemical system is shown in Figure 1. The mechanism of the system has been investigated sufficiently in previous study [22]. Under acidic conditions, both the electrochemical oxidation and Fenton reaction displays a great contribution to organic compounds degradation. A large amount of ·OH was produced at the BDD anode surface and Fenton reaction. Under alkaline conditions, the Fe^2+^ released from Fe_c_ electrode would coagulate and remove organic compounds from the solution by flocculation. At the same time, Fe(VI) was formed at alkaline conditions, which also has a great oxidation ability for contaminant degradation [22,27].

RSM was used to explore effects and interaction of different operating parameters in the BDD–Fe_c_–CF system for the electrochemical oxidation of landfill leachate after biological treatment. Current density, electrolytic time and pH value were selected as variables and their effects on the COD and NH_3_-N removal efficiency were investigated. Table 3 lists the COD and NH_3_-N removal efficiency in each case.

Under different operating conditions, the value of D_COD_ was in the range of 36.67~76.9% and that of D_NH3-N_ was in 36.89~91.2%. An empirical second-order polynomial model for predicting the optimal point was according to the following Equations (7) and (8):(7)$$Y_{COD}=14.26877+0.83805x_{1}+2.77004x_{2}-1.44441x_{3}+0.11x_{1}x_{2}-\;\;\;\;\;\;\;\;\;\;0.00075x_{1}x_{3}\;-0.041875x_{2}x_{3}-0.00157x_{1}{}^{2}-0.001575x_{2}{}^{2}+\;\;\;\;\;\;\;\;\;\;0.15583x_{3}{}^{2}$$
(8)$$Y_{NH3-N}=-32.69555+4.57275x_{1}+7.49271x_{2}-3.58828x_{3}+\;\;\;\;\;\;\;\;\;\;0.22367x_{1}x_{2}\;\;\;\;\;\;\;\;\;\;-0.00425x_{1}x_{3}+0.15229x_{2}x_{3}-0.09765x_{1}{}^{2}-\;\;\;\;\;\;\;\;\;0.55875x_{2}{}^{2}\;+0.26727x_{3}{}^{2}\;\;\;\;\;\;\;\;\;\;\;\;\;\;\;\;$$
where *x*_1_ is the normalized current density, *x*_2_ is the normalized electrolysis time and *x*_3_ is the normalized pH value.

The related statistical criteria provided by RSM was shown in Table 4. The correlation coefficients of R^2^ (0.9941 and 0.9768) and adjusted R^2^ (0.9865 and 0.9469) were high [28]. Predication R^2^ was calculated from predicted residual error sum of squares (PRESS) [29]. Adequate precision (AP) was a kind of signal-to-noise ratio which compared the range of the predicted values at the design points to the average prediction error. AP indicator should be 4 or more to be an appropriate model predictor [30,31,32]. The coefficient of variations (CV) of D_COD_ and D_NH3-N_ were 2.27% and 5.83% (CV < 10%), which represented a greater meticulousness and consistency of the model.

All the responses (Y) are as defined in text; R^2^: determination coefficient; Adj R^2^: adjusted R^2^; Pre R^2^: prediction R^2^; AP: adequate precision; SD: standard deviation; CV: coefficient of variation; PRESS: predicted residual error sum of squares.

The results of analysis of variance (ANOVA) for Equations (7) and (8) are shown in Appendix A. For both models, the F-values (130.62, 32.70) were larger than 0.05 and the probability values (<0.0001) were low, implying that the two models were significant. 

The residuals were considered to examine deviations between experimental and predicted results. A normal probability plot following a straight line was found in Figure 2a,b. In both subplots, the points or point clusters were located close to the diagonal line. Based on these results it can be seen that the errors were normally distributed and independent of each other [33]. Figure 2c,d shows excellent agreement between experimental and predicted results for D_COD_ and D_NH3-N_ in the BDD–Fe_c_–CF system in all cases. These results demonstrate that Equations (7) and (8) have high confidence for prediction of D_COD_ and D_NH3-N_.

According to the prediction results of Equations (7) and (8), the optimum conditions were a current density of 25 mA·cm^−2^, electrolytic time of 9 h and pH of 11. Under these operating conditions, a D_COD_ of 82.41% can be achieved with a D_NH3-N_ of 99.87%. The optimal conditions were tested three times and the actual average values of D_COD_ and D_NH3-N_ were 78.03% and 94.16%, respectively, which were very closed to the predicted ones. The relative standard deviation of of D_COD_ and D_NH3-N_ were 3.86% and 4.16%, both of which were acceptable. Effects of different operating parameters on D_COD_ and D_NH3-N_ in BDD–Fe_c_–CF system are demonstrated by three-dimensional response surface plots in Figure 3 and Figure 4, respectively. Electrolytic time was the most significant variable for D_COD_ and the affection order was electrolysis time > current density > pH. For D_NH3-N_, the affection of different variables was consistent with D_COD_.

As shown in Figure 3, electrolysis time and current density has positive effect on D_COD_. However, D_COD_ decreased first and then increased with pH increasing. This phenomenon meant D_COD_ was high in acidic and alkaline conditions but low in neutral conditions in the BDD–Fe_c_–CF system. Under acidic conditions, ·OH were formed by reaction between Fe^2+^ released from Fe_c_ electrode and H_2_O_2_ generated by dissolved oxygen reduction of the carbon felt cathode. This result was consistent with Meng, G. et al. [34], where the pH range of 2–4.5 was conducive for optimum COD removal in the electro-Fenton system. Under neutral conditions, the electrical conductivity reached its minimum value due to a more effective coagulation [3]. Under alkaline conditions, flocs were formed to remove organic compounds by coagulation as well as anode oxidation, electro-generated oxidants and Fe(VI) oxidation.

The effects of different operating parameters on D_NH3-N_ are shown in Figure 4. D_NH3-N_ increased along with electrolysis time and current density. The effect of pH on D_NH3-N_ was similar with on D_COD_. The removal efficiency was higher under acidic and alkaline conditions than neutral conditions. However, it should be noted that D_NH3-N_ under alkaline conditions was much higher than that under acidic conditions. The much higher D_NH3-N_ under alkaline conditions was probably due to the volatilization of NH_3_ (Equation (9)).
NH_4_^+^ + OH^−^ → NH_3_ ↑ + H_2_O(9)

Moreover, Cl^−^ with high concentrations in landfill leachate also has a significant effect on D_COD_ and D_NH3-N_ during the electrochemical oxidation process. The existing form of active chlorine in electrolyte was mainly affected by pH value. Under acidic conditions (pH < 7.5), chlorine existed as HOCl (E^θ^ = 1.63 V), while under alkaline conditions (pH > 7.5), OCl^−^ (E^θ^ = 0.90 V) was the domain ion in electrolyte (Equations (10)–(12)). Therefore, under acidic conditions, high D_COD_ and D_NH3-N_ was also due to the stronger oxidation ability of HOCl (Equations (13) and (14)) [35,36].
2 Cl^−^ → Cl_2_ + 2 e^−^(10)
Cl_2_ + H_2_O → HOCl + H^+^ + Cl^−^(11)
Cl_2_ + OH^−^ → OCl^−^+ Cl^−^+ H_2_O(12)
2/3 NH_4_^+^ + HOCl → 1/3 N_2_ + H_2_O + 5/3 H^+^ + Cl^−^(13)
NH_4_^+^ + 4 HOCl → NO_3_^−^ + H_2_O + 6 H^+^ + 4 Cl^−^(14)

### 3.2. Analyzed by GC–MS

The results of the GC–MS analysis of landfill leachate before and after electrochemical treatment under the optimal reaction conditions (current density of 25 mA·cm^−2^, electrolytic time of 9 h, pH value of 11) are shown in Figure 5. At least 56 types of organic compounds were found in landfill leachate before treatment, including alkanes and olefins, alcohols, aldehyde and ketones, amides and nitrile, aromatic hydrocarbon, carboxylic acids, esters, heterocyclic compounds and hydroxybenzenes (Appendix A). In terms of molecular structure, most of them were long-chain or circular organics with complex structures. Some of the compounds, such as chlorobenzene, 1,2,4-trichlorobenzene, 1,1-dichloroethylene, and N-nitrosodimethylamine, are priority pollutants defined by US EPA [37] and were also observed in other leachates by Scandelai et al. [38]. After electrochemical treatment, only 16 types of organic compounds were left and some of them were new in the effluents of the electrochemical treatment process. This could indicate that the refractory organic contaminants were degraded and converted into small molecules and, thus, the proposed process has a good destructive effect on long-chain alkanes compared with the traditional electro-Fenton system [34]. This is due to the effluents from BDD–Fe_c_–CF system mainly containing small organic compounds, which were intermediate products formed by the oxidation, such as methylene chloride, chloromethane sulfonyl chloride, chloroform and so on (Appendix A). 

## 4. Conclusions

Landfill leachate after biological treatment processes was treated in BDD–Fe_c_–CF system and the RSM was applied to analyze the effects of the operating parameters of the current density, electrolysis time and pH value. This system can remove COD and NH_3_-N efficiently in the pH range of 3–11. Results showed that the order of different operating parameters was electrolytic time > current density > pH. Under the optimum conditions (current density of 25 mA·cm^−2^, electrolytic time of 9 h, pH value of 11), D_COD_ and D_NH3-N_ can achieve 81.3% and 99.8%, respectively. The predicted values calculated with the model equations were very close to the experimental values and the models were highly significant. Among the three operating parameters, both electrolysis time and current density have positive effects on D_COD_ and D_NH3-N_. However, the effect of pH is complicated. D_COD_ and D_NH3-N_ were high under acidic and alkaline conditions while low under neutral conditions. This phenomenon was mainly due to the mechanism of the BDD–Fe_c_–CF system, which produced ·OH and HOCl under acidic conditions and formed flocs composed of Fe(OH)_2_ and Fe(OH)_3_, which were formed to remove organic compounds by coagulation, Fe(VI) oxidation and OCl^−^ under alkaline conditions, along with volatilization of NH_3_. GC–MS analyzation showed that electrochemical oxidation removed organic compounds efficiently for landfill leachate after biological treatment processes with great potential for practical application.

## Figures and Tables

**Figure 1 ijerph-19-07745-f001:**
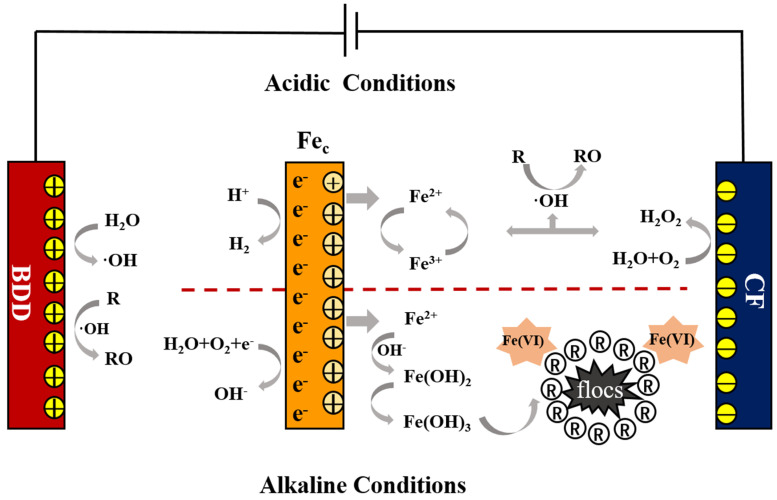
Schematic diagram for mechanism of BDD–Fe_c_–CF system.

**Figure 2 ijerph-19-07745-f002:**
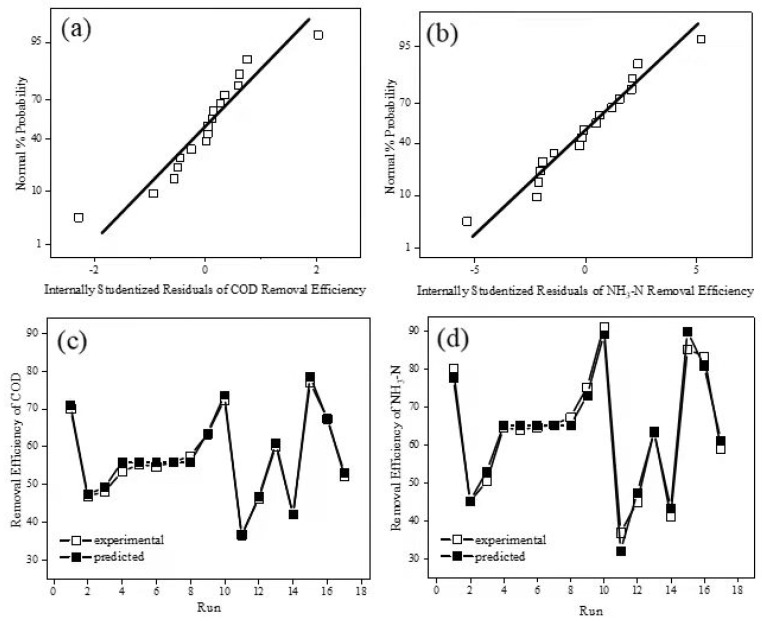
Normal probability plot of the internally studentized residuals for COD (**a**) and NH_3_−N (**b**) removal. Predicted versus actual values plot for COD (**c**) and NH_3_−N (**d**) removal.

**Figure 3 ijerph-19-07745-f003:**
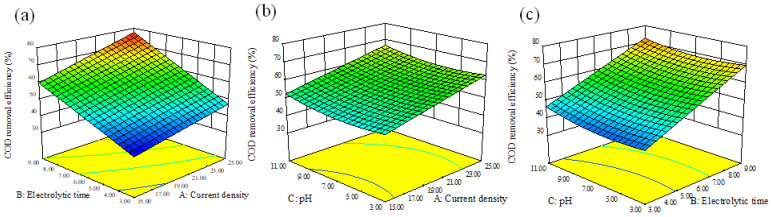
Response surface plots of the combined effects of (**a**) current density and electrolysis time, (**b**) current density and pH, (**c**) electrolysis times and pH on COD removal efficiency.

**Figure 4 ijerph-19-07745-f004:**
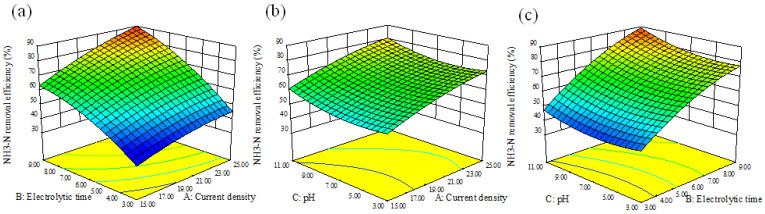
Response surface plots of the combined effects of (**a**) current density and electrolysis time, (**b**) current density and pH, (**c**) electrolysis times and pH on NH_3_-N removal efficiency.

**Figure 5 ijerph-19-07745-f005:**
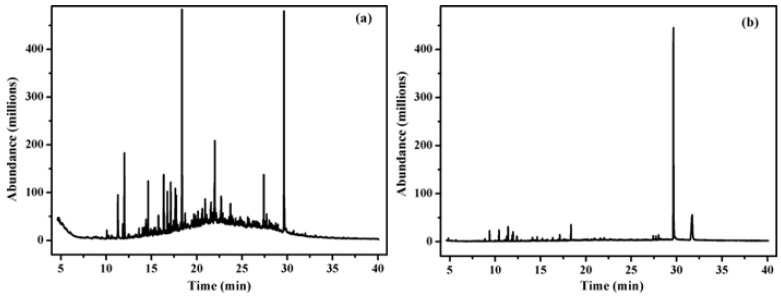
GC–MS analysis for organics in the influent (**a**) and effluent (**b**) of the electrochemical reactor.

**Table 1 ijerph-19-07745-t001:** Character of landfill leachate samples.

Parameters	Unit	Value (Range)	Value (Average)
pH	-	6.85~7.39	7.22
COD	mg·L^−1^	2163~2602	2464
NH_3_-N	mg·L^−1^	139.5~178.2	154.4

**Table 2 ijerph-19-07745-t002:** Level and code of experimental variables based on BBD.

Variables	Symbol	Units	Codes and Levels		
			−1	0	1
Current density	*x* _1_	mA·cm^−2^	15	20	25
Electrolysis time	*x* _2_	h	3	6	9
pH value	*x* _3_	-	3	7	11

**Table 3 ijerph-19-07745-t003:** Design matrix and the experimental responses.

Number	Current Density (mA·cm^−2^)	Electrolytic Time (h)	pH	D_COD_ (%)	D_NH3-N_ (%)
1	20	9.00	3.00	70.01	80.14
2	25	3.00	7.00	46.8	45.09
3	15	6.00	3.00	48.09	50.42
4	20	6.00	7.00	53.29	64.53
5	20	6.00	7.00	55.24	63.98
6	20	6.00	7.00	54.7	64.7
7	20	6.00	7.00	55.83	65.29
8	20	6.00	7.00	57.52	67.3
9	25	6.00	3.00	63.29	75.26
10	20	9.00	11.00	72.38	91.2
11	15	3.00	7.00	36.67	36.89
12	20	3.00	11.00	46.33	45.02
13	15	9.00	7.00	60.17	63.58
14	20	3.00	3.00	41.95	41.27
15	25	9.00	7.00	76.9	85.2
16	25	6.00	11.00	67.42	83.4
17	15	6.00	11.00	52.28	58.9

**Table 4 ijerph-19-07745-t004:** The related statistical criteria values of models.

Model	R^2^	Adj R^2^	Pre R^2^	AP	SD	CV(%)	PRESS
Y_1_	0.9941	0.9865	0.9769	42.000	1.28	2.27	44.69
Y_2_	0.9768	0.95469	0.6513	20.305	3.71	5.83	1445.05

## Data Availability

Not applicable.

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
