# Peer review of "Electrochemical Oxidation of Landfill Leachate after Biological Treatment by Electro-Fenton System with Corroding Electrode of Iron"

_ijerph, 2022, doi:10.3390/ijerph19137745_

Round 1

Reviewer 1 Report

1. The manuscript needs improvement in terms of results and discussions. You are advised to relate your research with recent uses of Fenton technology in terms of similarities and differences. 

2. pH effects on chemicals are very complex. How can your summarize your acidic and basic pH conditions in your developed system. Give answer with reference to recent advancements.

Author Response

Specific Comments:

  1. The manuscript needs improvement in terms of results and discussions. You are advised to relate your research with recent uses of Fenton technology in terms of similarities and differences. 

Response: Thank you for your valuable suggestions. The results and discussions have been modified and highlighted in the revised manuscript.

P4, L167-176: “The scheme of BDD-Fec-CF electrochemical system has been shown in Fig. 1. Mechanism of the system has been investigated sufficiently in previous study [23]. Under acidic conditions, both electrochemical oxidation and Fenton reaction has great contribution for organic compounds degradation. Large amount of ·OH was produced at BDD anode surface and Fenton reaction. Under alkaline conditions, Fe2+ released from Fec electrode would coagulate and remove organic compounds from solution by flocculation. At the same time, Fe(Ⅵ) was formed at alkaline conditions which also has great oxidation ability for contaminant degradation [23, 28].”

P8, L239-249: “As shown in Fig. 3, electrolysis time and current density has positive effect on DCOD. However, DCOD decreased first and then increased with pH increasing. This phenomenon meant DCOD was high at acidic and alkaline conditions while it was low at neutral conditions in BDD-Fec-CF system. Under acidic conditions, ·OH were formed by reaction between Fe2+ released from Fec electrode and H2O2 generated by dissolved oxygen reduction at carbon felt cathode. This result was consistent with Meng, G. et al. [35] that at the pH range of 2-4.5 is conducive for optimum COD removal in electro-Fenton system. Under neutral conditions, the electrical conductivity reached its minimum value due to a more effective coagulation [36]. Under alkaline conditions, flocs were formed to remove organic compounds by coagulation besides anode oxidation, electro-generated oxidants and Fe(VI) oxidation.”

P9, L268-285: “The results of the GC–MS analysis of landfill leachate before and after electrochemical treatment under the optimal reaction conditions (current density of 25 mA·cm−2, electrolytic time of 9 h, pH value of 11) are shown in Fig. 5. At least 56 types of organic compounds was found in landfill leachate before treatment, including alkanes and olefins, alcohols, aldehyde and ketones, amides and nitrile, aromatic hydrocarbon, carboxylic acids, esters, heterocyclic compounds and hydroxybenzenes (Table S3). In terms of molecular structure, most of them were long-chain or circular organics with complex structures. Among of the compounds, such as chlorobenzene, 1,2,4-trichlorobenzene, 1,1-dichloroethylene, N-nitrosodimethylamine are priority pollutants defined by US EPA [39] and were also observed in others leachate by Scandelai et al. [40]. After electrochemical treatment, only 16 types organic compounds were left and some of them were new in effluent of electrochemical treatment process. It could be indicated that the refractory organic contaminants were degraded and converted into small molecules and has a good destructive effect on long-chain alkanes compared with traditional electro-Fenton system [41]. This is due to the effluent from BDD-Fec-CF system mainly contained some small organic compounds which were intermediate products formed from the oxidation, such as Methylene Chloride, Chloromethane sulfonyl chloride, and Chloroform and so on (Table S4). ”

  1. pH effects on chemicals are very complex. How can your summarize your acidic and basic pH conditions in your developed system. Give answer with reference to recent advancements.

Response: Thank you for your valuable suggestions. It is true that pH value is important and complex for organic compounds degradation, especially in Fenton system. In traditional Electro-Fenton reaction, the pH value of the solution for this method is very narrow, usually should be controlled at 3. When pH value increased, Fe(III) oxyhydroxides were formed with a large amount of iron mud produced. In previous study, a novel way to introduce Fe2+ by corroding electrode of Fec in BDD anode system was established (BDD-Fec-CF) [1]. Performance of the novel Electro-Fenton system under different pH conditions was investigated. Results showed that under acidic (pH was 3), neutral (pH was 5) and alkaline conditions (pH was 11), p-nitrophenol has high removal efficiency in the Electro-Fenton system. Under acidic condition, Electro-Fenton reaction between Fe2+ released from Fec electrode and H2O2 generated at CF cathode enhanced the oxidation ability. Under alkaline condition, flocs composed by Fe(OH)2 and Fe(OH)3 were formed to remove organic compounds by coagulation. Joint action of anodic oxidation, electro-generated oxidants oxidation, Electro-Fenton reaction under acidic conditions, as well as coagulation and Fe(VI)-oxidation under alkaline conditions has improved organic pollutants degradation in the BDD-Fec-CF system.

Since BDD-Fec-CF system has high efficiency in a wide pH range of 3-11, it was applied for advanced treatment of landfill leachate after biological treatment process. During the present study, pH value was set to be 3, 7 and 11 to represented acidic, neutral and alkaline conditions.

In other research on landfill leachate treatment by Fenton reaction, pH is also important and usually controlled in acidic condition (pH was 2-4.5) [2, 3]. According to the report by Ghahrchi M. et al. [4], at the range of neutral pH, most of Fe3+ ions formed through the electrochemical process generate iron hydroxide flakes and the electrical conductivity reached its minimum value, probably due to a more effective coagulation and reduction in the ions present in the leachate. In this study, both DCOD and DNH3-N in pH range of 3-11 can achieve high efficiency, which demonstrate its great potential for practical application.

References

[1] Li, H.; Xing, X.; Wang, K.; Zhu, X.; Jiang, Y.; Xia, J. Improved BDD anode system in electrochemical degradation of p-nitrophenol by corroding electrode of iron. Electrochim. Acta. 2018, 291: 335-342. [ScienceDirect]

[2] Mahtab, M. S., Islam, D. T., Farooqi, I. H. Optimization of the process variables for landfill leachate treatment using Fenton based advanced oxidation technique. Eng. Sci. Technol.2021, 24(2): 428-435. [ScienceDirect]

[3] Meng, G., Jiang, N., Wang, Y., Zhang, H., Tang, Y., Lv, Y., Bai, J. Treatment of coking wastewater in a heterogeneous electro-Fenton system: Optimization of treat ment parameters, characterization, and removal mechanism. J. Water Process Eng. 2022, 45: 102482. [ScienceDirect]

[4] Ghahrchi M.; Rezaee A. Electro-catalytic ozonation for improving the biodegradeability of mature landfill leachate. J. Environ. Manage. 2020, 254: 109811. [PubMed]

Reviewer 2 Report

In this manuscript, the authors designed a BDD-Fec-CF system to treat landfill leachate, analyzed the influence of operation parameters with RSM, and obtained the best operation conditions. This method can effectively remove organic matter. It seems interesting and significant. However, there are some problems. I recommend this paper can be published after minor revision.

1. It seems that the font of first Introduction paragraph is different from other paragraphs. Please check it carefully.

2. Page 2, line 64. There was a mistake in the twelfth reference.

3. Page 4, line 139. The preparation method of GC-MS analytical samples should be followed by references. Page 9, line 268. The reference should be added after “and were also observed in other leachate by Scandelai et al.”

4. It seems that the font and format of Table 2 are inconsistent.

5. All numbers in DNH3-N in the article are not changed to Subscripts such as lines 219 and 220 on page 7. 

6. Can you explain why the matrix and experimental response are designed as shown in Table 3? It seems a little unreasonable. Maybe it can be listed in order by fixing two variables and changing another variable, which will be more intuitive.

Author Response

Specific Comments:

  1. It seems that the font of first Introduction paragraph is different from other paragraphs. Please check it carefully.

Response: Thank you for your valuable suggestions. The font of first Introduction paragraph has been modification in the revised manuscript (P1/L31~48).

  1. Page 2, line 64. There was a mistake in the twelfth reference.

Response: Thank you for your valuable suggestions. The mistake has been corrected in the revised manuscript.

P2/L65: “Among all kinds of anode materials, boron-doped diamond (BDD) anode with its high stability, wide potential window and extremely high oxygen evolution properties has been regarded as one of the best electrodes [12-17].”

[12] Michele, M.; Vacca, A.; Palmas, S. Electrochemical treatment as a pre-oxidative step for algae removal using Chlorella vulgaris as a model organism and BDD anodes, Chem. Eng. J. 2013, 219: 512-519. [ScienceDirect]

  1. Page 4, line 139. The preparation method of GC-MS analytical samples should be followed by references. Page 9, line 268. The reference should be added after “and were also observed in other leachate by Scandelai et al.”

Response: Thank you for your valuable suggestions. Two references have been added in the revised manuscript (Reference [27] and [40]).

P4, L139: “Samples for GC-MS analysis were prepared by following the procedure reported by Lei et al. [27].”

P9, L277: “Among of the compounds, such as chlorobenzene, 1,2,4-trichlorobenzene, 1,1-dichloroethylene, N-nitrosodimethylamine are priority pollutants defined by US EPA [39] and were also observed in others leachate by Scandelai et al. [40].”

[27] Lei, Y.; Shen, Z.; Huang, R.; Wang, W. Treatment of landfill leachate by combined aged-refuse bioreactor and electro-oxidation. Water Res. 2007, 41: 2417-2426. [PubMed]

[40] Scandelai, A. P. J.; Zotesso, J. P.; Jegatheesan, V.; Cardozo-Filho, L.; Tavares, C. R. G. Intensification of supercritical water oxidation (ScWO) process for landfill leachate treatment through ion exchange with zeolite. Waste Manage. 2020, 101: 259-267. [ScienceDirect]

  1. It seems that the font and format of Table 2 are inconsistent.

Response: Thank you for your valuable suggestions. The font and format of Table 2 has been modified in revised manuscript.

Table 2. Level and code of experimental variables based on BBD.

Variables

Symbol

Units

Codes and levels

-1

0

1

Current density

x1

mA·cm-2

15

20

25

Electrolysis time

x2

h

3

6

9

pH value

x3

-

3

7

11

  1. All numbers in DNH3-N in the article are not changed to Subscripts such as lines 219 and 220 on page 7. 

Response: Thank you for your valuable suggestions. All the “DNH3-N” has been corrected into “DNH3-N” and highlighted in the revised manuscript.

  1. Can you explain why the matrix and experimental response are designed as shown in Table 3? It seems a little unreasonable. Maybe it can be listed in order by fixing two variables and changing another variable, which will be more intuitive.

Response: Thank you for your valuable question. In the present study, Response surface methodology (RSM) with Box-Behnken Design (BBD) model was applied to determine the interaction of different operating variables for COD and NH3-N removal.

RSM is a sophisticated statistical experimental design tool used for optimization of different processes. In the optimization of electrochemical oxidation of landfill leachate, the objective was to optimize the response surface which get influenced by different parameters as well as to identify the relationship between variable factors and their responses [1].

BBD is efficient and easy to arrange and explain the experimental findings compared to control-variables method [2]. Usually, one-parameter-at-a-time procedure required spending of longer time and abundant tests for determining optimum levels. BBD method design does not require the installation of the axial point and abundant continuous tests and the number of factors can be set to 3–7, which makes it popular in optimizing operational parameters.

BBD is implemented by Design-Expert software (8.0.6). The total number of runs required by BBD is defined by N = 2K(K-1) + C0, where K is the number of variables studied 3 and C0 is the number of central points 5. For present BBD with RSM study, 17 runs were carried out along with 5 replicates at the center of the design in order to excess pure error.

References

[1] Khoshnamvand, N., Kord Mostafapour, F., Mohammadi, A., Faraji, M. Response surface methodology (RSM) modeling to improve removal of ciprofloxacin from aqueous solutions in photocatalytic process using copper oxide nanoparticles (CuO/UV). AMB Express, 2018, 8(1): 1-9. [PubMed]

[2] Li, R., Yang, J., Pan, J., Zhang, L., Qin, W. Effect of immobilization on growth and organics removal of chlorella in fracturing flowback fluids treatment. J. Environ. Manage. 2018, 226:163-168. [ScienceDirect]

Reviewer 3 Report

Tang et al. Electrochemical Oxidation of Landfill leachate after Biological Treatment by Electro-Fenton System with Corroding Electrode of Iron. The topic of the manuscript is relevant and suitable for publishing in International Journal of Environmental Research and Public Health. The authors need to correct several main issues before publication and my comments are given as below.

1. Table 1, only pH, COD and NH3-N were detected in landfill leachate samples? Other parameters need added.

2. Lines 261-272, the changes of organics during treatment process need discussed deeply.

3. The section of conclusions should be improved.

Author Response

Specific Comments:

  1. Table 1, only pH, COD and NH3-N were detected in landfill leachate samples? Other parameters need added.

Response: Thank you for your valuable suggestions. It is true that other parameters are important for landfill leachate such as conductivity, concentration of Cl- and BOD5. However, the water samples were taken after biological treatment process for advanced oxidation and we focused on COD and NH3-N removal efficiency. Therefore, only COD, NH3-N and pH, which is important for Electro-Fenton reaction, were selected and analyzed in the present study. In the following research, we will supplement the other parameters and make further investigation for landfill leachate treatment. Many thanks for your kindly suggestion again.

  1. Lines 261-272, the changes of organics during treatment process need discussed deeply.

Response: Thank you for your valuable suggestions. This section has been rewritten and highlighted in the revised manuscript. The structure of organic compounds before and after the reaction is described and combined with the recent progress of Electro-Fenton.

P9, L268-285: “The results of the GC–MS analysis of landfill leachate before and after electrochemical treatment under the optimal reaction conditions (current density of 25 mA·cm−2, electrolytic time of 9 h, pH value of 11) are shown in Fig. 5. At least 56 types of organic compounds was found in landfill leachate before treatment, including alkanes and olefins, alcohols, aldehyde and ketones, amides and nitrile, aromatic hydrocarbon, carboxylic acids, esters, heterocyclic compounds and hydroxybenzenes (Table S3). In terms of molecular structure, most of them were long-chain or circular organics with complex structures. Among of the compounds, such as chlorobenzene, 1,2,4-trichlorobenzene, 1,1-dichloroethylene, N-nitrosodimethylamine are priority pollutants defined by US EPA [39] and were also observed in others leachate by Scandelai et al. [40]. After electrochemical treatment, only 16 types organic compounds were left and some of them were new in effluent of electrochemical treatment process. It could be indicated that the refractory organic contaminants were degraded and converted into small molecules and has a good destructive effect on long-chain alkanes compared with traditional electro-Fenton system [41]. This is due to the effluent from BDD-Fec-CF system mainly contained some small organic compounds which were intermediate products formed from the oxidation, such as Methylene Chloride, Chloromethane sulfonyl chloride, and Chloroform and so on (Table S4)”

  1. The section of conclusions should be improved.

Response: Thank you for your valuable suggestions. The conclusions have been rewritten and highlighted in the revised manuscript.

P9, L289~L306: “Landfill leachate after biological treatment process was treated in BDD-Fec-CF system and RSM was applied to analyze the effects of operating parameters of current density, electrolysis time and pH value. This system can remove COD and NH3-N efficiently in the pH range of 3-11. Results showed that the order of different operating parameters was electrolytic time > current density > pH. Under the optimum conditions (current density of 25 mA·cm−2, electrolytic time of 9 h, pH value of 11), DCOD and DNH3-N can be achieved 81.3% and 99.8%, respectively. The predicted values calculated with the model equations were very close to the experimental values and the models were highly significant. Among the three operating parameters, both electrolysis time and current density have positive effect on DCOD and DNH3-N. However, effect of pH was complicated. DCOD and DNH3-N was high under acidic and alkaline conditions while low under neutral conditions. This phenomenon was mainly due to the mechanism of BDD-Fec-CF system which produced ·OH and HOCl under acidic conditions and formed flocs composed by Fe(OH)2 and Fe(OH)3 were formed to remove organic compounds by coagulation, Fe(Ⅵ) oxidation and OClunder alkaline conditions along with volatilization of NH3. GC-MS analyzation showed that electrochemical oxidation removed organic compounds efficiently for landfill leachate after biological treatment process with great potential for practical application.”

Reviewer 4 Report

The authors have written a well-structured article. The authors did a lot of work of experimental design, data analysis, and the results presenting. However, some revisions are still needed before accepting:

1) Please check the references in the introduction (page 2, line 64),

2) In abstract and in introduction please write what the abbreviations COD, BOD mean,

3) Pages 4/5 -  please shortly (at least 1 sentence) elaborate, what the Authors meant when they wrote “Under acidic conditions, both electrochemical oxidation and Fenton reaction has great contribution for organic compounds degradation. Under alkaline conditions, coagulation and Fe() oxidation affected organic compounds removal.” What was the effect on the removal of organic compounds?

4) Page 4:  2.4. Response surface methodology

“RSM was utilized to optimize the operating conditions and analyze the effects of different operating parameters on electrochemical treatment of landfill leachate taken from biological treatment process.” What different parameters are you referring to? Please list them.

5) Page 8: 3.2. Analyzed by GC-MS,

Some of the compounds are priority pollutants defined by US EPA [35] and were also observed in other leachate by Scandelai et al.. - citation number missing, additionally please list in the text examples of priority pollutants defined by US EPA. 

Author Response

Specific Comments:

  1. Please check the references in the introduction (page 2, line 64)

Response: Thank you for your valuable suggestions. The mistake has been modified in the revised manuscript.

P2, L65: “Among all kinds of anode materials, boron-doped diamond (BDD) anode with its high stability, wide potential window and extremely high oxygen evolution properties has been regarded as one of the best electrodes [12-17].”

[12] Michele, M.; Vacca, A.; Palmas, S. Electrochemical treatment as a pre-oxidative step for algae removal using Chlorella vulgaris as a model organism and BDD anodes, Chem. Eng. J. 2013, 219: 512-519. [ScienceDirect]

  1. In abstract and in introduction please write what the abbreviations COD, BOD mean.

Response: Thank you for your valuable suggestions. The meanings of COD and BOD appearing for the first time in the abstract and introduction sections have been added in the revised manuscript.

P1, L15: Abstract: “Effects of variables including current density, electrolytic time and pH on chemical oxygen demand (COD) and ammonia nitrogen (NH3-N) removal efficiency were analyzed.”

P2, L41-L43: Introduction: “The leachate contains large amount of dissolved organic compounds, hazardous chemicals, soluble salts, heavy metals with high chemical oxygen demand (COD) and ammonia nitrogen (NH3-N) concentration and low biochemical oxygen demand/chemical oxygen demand (BOD/COD) ratio.”

  1. Pages 4/5 - please shortly (at least 1 sentence) elaborate, what the Authors meant when they wrote “Under acidic conditions, both electrochemical oxidation and Fenton reaction has great contribution for organic compounds degradation. Under alkaline conditions, coagulation and Fe(Ⅵ) oxidation affected organic compounds removal.” What was the effect on the removal of organic compounds?

Response: Thank you for your valuable suggestions. This section has been rewritten and the specific effect of the removal of organic compounds under acidic and alkaline conditions is briefly explained.

P5, L172-176: “Large amount of ·OH was produced at BDD anode surface and Fenton reaction. Under alkaline conditions, Fe2+ released from Fec electrode would coagulate and remove organic compounds from solution by flocculation. At the same time, Fe(Ⅵ) was formed at alkaline conditions which also has great oxidation ability for contaminant degradation [1, 2].”

[1] Bakraouy, H., Souabi, S., Digua, K., Dkhissi, O., Sabar, M., Fadil, M. Optimization of the treatment of an anaerobic pretreated landfill leachate by a coagulation–flocculation process using experimental design methodology. Process Saf. Environ. 2017, 109: 621-630.  [ScienceDirect]

[2] Li, H.; Xing, X.; Wang, K.; Zhu, X.; Jiang, Y.; Xia, J. Improved BDD anode system in electrochemical degradation of p-nitrophenol by corroding electrode of iron. Electrochim. Acta. 2018, 291: 335-342. [ScienceDirect]

  1. Page 4:  2.4. Response surface methodology. “RSM was utilized to optimize the operating conditions and analyze the effects of different operating parameters on electrochemical treatment of landfill leachate taken from biological treatment process.” What different parameters are you referring to? Please list them.

Response: Thank you for your valuable suggestions. In this work, the current density (X1), electrolytic time (X2) and pH value (X3) were selected as different parameters.

  1. Page 8: 3.2. Analyzed by GC-MS. Some of the compounds are priority pollutants defined by US EPA [35] and were also observed in other leachate by Scandelai et al.. - citation number missing, additionally please list in the text examples of priority pollutants defined by US EPA. 

Response: Thank you for your valuable suggestions. The reference (Reference [40]) has been added and the examples of priority pollutants defined by US EPA have been list in the revised manuscript.

P9, L276-278: “Among of the compounds, such as chlorobenzene, 1,2,4-trichlorobenzene, 1,1-dichloroethylene, N-nitrosodimethylamine are priority pollutants defined by US EPA [39] and were also observed in others leachate by Scandelai et al. [40].”

[40] Scandelai, A. P. J.; Zotesso, J. P.; Jegatheesan, V.; Cardozo-Filho, L.; Tavares, C. R. G. Intensification of supercritical water oxidation (ScWO) process for landfill leachate treatment through ion exchange with zeolite. Waste Manage. 2020, 101: 259-267. [ScienceDirect]

Round 2

Reviewer 1 Report

I accept the answers of the authors and the manuscript can be published.